# A Rare Case of Upper Gastrointestinal Bleeding: Osler-Weber-Rendu Syndrome

**DOI:** 10.3390/medicina58030333

**Published:** 2022-02-22

**Authors:** Anna Jargielo, Anna Rycyk, Beata Kasztelan-Szczerbinska, Halina Cichoz-Lach

**Affiliations:** 1Banacha Campus, Medical University of Warsaw, Zwirki i Wigury 61 St., 02-091 Warsaw, Poland; ania.jargielo@gmail.com; 2Department of Gastroenterology with Endoscopy Unit, Medical University of Lublin, Jaczewskiego 8 St., 20-954 Lublin, Poland; beata.szczerbinska@op.pl (B.K.-S.); lach.halina@wp.pl (H.C.-L.)

**Keywords:** Osler-Weber-Rendu disease, hereditary hemorrhagic telangiectasia (HHT), mucocutaneous telangiectasias, arterio-venous malformations, vascular malformations (VMs), Curaçao criteria, upper gastrointestinal bleeding

## Abstract

Osler-Weber-Rendu disease, also known as hereditary hemorrhagic telangiectasia (HHT), is a rare, autosomal dominant condition that affects approximately 1 in 5000 patients causing abnormal blood vessel formation. HHT patients have mucocutaneous telangiectasias and arteriovenous malformations in various organs. The most prominent symptom of HHT is epistaxis, which, together with gastrointestinal bleeding, may cause iron deficiency anemia. This study is a case report of a 62-year-old patient who was admitted to the Department of Gastroenterology due to acute upper gastrointestinal bleeding and a history of recurrent epistaxis and melena for 4 days, which was confirmed in digital rectal examination. Urgent upper gastrointestinal endoscopy revealed active bleeding from multiple angioectatic spots with bright-looking salmon-colored patches in the antrum and the body suggestive of HHT. The bleeding from two angioectatic spots was stopped by argon plasma coagulation, and four clips were placed to provide good hemostasis. The patient was treated with a proton pomp inhibitor infusion and iron infusion. She was discharged with no signs of GI bleeding, normalized iron levels and a diagnosis of HHT. She was referred to further genetic testing, including evaluation of first-degree relatives. She also had performed unenhanced thin-cut computed tomography (CT) with angiography to exclude the presence of pulmonary arteriovenous malformations (PAVMs). Due to the fact that the patient did not manifest any other HHT-related symptoms and that the instrumental screening discloses no silent AVMs in other organs, the “watch-and-wait strategy” was applied. Although, Osler-Weber-Rendu syndrome is widely described in the medical literature, effective treatment of gastrointestinal telangiectasias is not always available and still lacks standardization to date, which makes the management of gastroenterological involvement still a challenging issue.

## 1. Introduction

Osler-Weber-Rendu disease, also known as hereditary hemorrhagic telangiectasia (HHT), is a rare, autosomal dominant disorder characterized by multiple vascular lesions ranging from mucocutaneous telangiectasias to larger arteriovenous malformations. Heterozygous pathogenic mutations in *ENG*, *ACVRL1/ALK1* and *MADH4/SMAD4* genes account for the disease. Large malformations mostly occur in the lungs, liver, brain, gastrointestinal tract and rarely in the pancreas [1]. In case of appearance of onset of telangiectasias in gastrointestinal tract, bleeding and subsequent iron deficiency anemia can arise, which require appropriate diagnostic work-up and clinical management. However, the most frequent HHT-related manifestation is represented by repeated epistaxis episodes, secondary to telangiectasia of the nasal mucosa. Furthermore, patients with HHT can suffer from cerebral abscess and/or cerebral infarctions, which could be prevented by screening and proper care. Several cases of gastrointestinal bleeding are a life-threatening emergency; therefore, it is crucial to establish the diagnosis and conduct a proper screening in order to prevent possible complications. Assessing a proper diagnosis of HHT in presence of otherwise unexplained GI is crucial to set up proper clinical instrumental survey of silent AVMs potentially affecting organs other than the GI tract, leading to other complications. Physicians should be aware of rare etiologies of gastrointestinal bleeding such as HHT. Some studies showed that in untreated populations of HHT patients, the median life expectancy was lower than in patients without HHT [2].

## 2. HHT Diagnosis

The diagnosis is based on the Curaçao criteria from 1999 [3,4]. HHT is deemed probable if patients present two criteria and definitive if three out of four criteria are met. If none or only one criterion is met, this makes the diagnosis unlikely [5,6]. The criteria are described in Table 1.

The diagnosis may be confirmed through a molecular gene test identifying a pathogenic sequence variant in *ENG, ACVRL1* or *SMAD4*. In this way, the HHT subtype can be identified and first-degree relatives can be tested for the disease. HHT is estimated to affect approximately 1 in 5000–6000 Europeans [7,8]. More than 76–80% of all cases of HHT are due to mutations in either *ENG* or *ACVRL1* [9,10]. HHT-related GI bleeding develops in approximately 13–30% of HHT patients, typically manifesting in the fifth and sixth decades [10,11,12]. According to Lesca et al., HHT patients with gastrointestinal bleeding display a mutation in *ACVRL1* more frequently than in the *ENG* gene, according to most genotype–phenotype correlation studies performed thus far [13]. Meanwhile, other researchers find no correlation between GI bleeding and genotype [14,15].

## 3. Case Report

Here in, we present a case of a 62-year-old female who was admitted to the Department of Gastroenterology with a suspicion of acute upper gastrointestinal bleeding. Two years ago, she underwent a surgical excision of a lentigo maligna of the interscapular region. She has had a history of recurrent epistaxis since the age of 11. On the day of the admission to hospital, the patient had melena for the first time in her life, lasting 4 days before the admission, which was confirmed in digital rectal examination. However, she had also received iron supplementation due to anemia before visiting the hospital. On admission, she was hemodynamically stable. Physical examination of the abdomen revealed no tenderness. The patient had telangiectatic lesions at characteristic sites on oral mucosa and previously she had telangiectatic lesions on her lips. The laboratory work-up at admission revealed the following: hemoglobin of 9.0 g/dl, hematocrit of 29.4%, serum ferritin of 7.0 ng/mL, serum iron of 10.0 μg/dL, white blood cells of 3.42 × 10^3^/μL and platelet count of 241 × 10^3^/μL. The coagulation profile was normal. The patient’s glycosylated hemoglobin, vitamin B_12_, folate levels and serum electrolytes were within normal range. The renal parameters and liver function tests were normal. Table 2 presents laboratory test results in our patient.

The patient underwent urgent upper gastrointestinal endoscopy, which revealed active bleeding from multiple angioectatic spots with bright-looking salmon-colored patches in the antrum and the body suggestive of HHT. The bleeding from two angioectatic spots was stopped by argon plasma coagulation, and four clips were placed to provide good hemostasis (Figure 1A–E). 

The patient was treated with a proton pomp inhibitor infusion and iron infusion. She was discharged with no signs of GI bleeding, normalized anemia parameters and diagnosis of HHT. A family history of epistaxis or any other potential HHT manifestation is unknown. In order to identify the underlying genetic mutation and enroll first-degree relatives to further genetic testing, the patient was referred to mutational analysis of the HHT-causing genes. Due to the fact that the patient did not manifest any other HHT-related symptoms and that the instrumental screening discloses no silent AVMs in other organs, the “watch-and-wait strategy” was applied. Our patient seems to represent an HHT case displaying diagnostic delay, a phenomenon typically occurring in Osler-Weber-Rendu syndrome [16], as the diagnosis was established after 11 years of unexplained nosebleeds. After a six-month control of the patient’s situation, it has been found that she has experienced two episodes of epistaxis treated symptomatically (tranexamic acid 500 mg 3 times a day).

## 4. Discussion

Osler-Weber-Rendu syndrome is named after Sir William Osler (Canadian physician), Frederick Parkes Weber (English dermatologist) and Henri Jules Louis Marie Rendu (French dermatologist) who independently described the condition on the cusp of the 19th and 20th centuries [17]. In June 1999, the Curaçao criteria were established to diagnose this disease [3].

Nowadays, while diagnosing HHT, some difficulties may be encountered as it is further influenced by socioeconomic and geographical factors (a higher incidence in Afro-Caribbean residents of Curaçao and Bonaire) [2,6]. Nevertheless, as the awareness of this condition rose since the moment it had been discovered and described in the literature, the reported proportion of HHT cases rose as well. 

In 2016, six countries set up a working group dedicated to HHT within what became the European Reference Network on Rare Multisystemic Vascular Diseases [18]. 

Many HHT-related manifestations become apparent in adulthood, at a variable onset age, due to the age-dependent expressivity of the disease. However, at a lesser frequency, some manifestations can also occur in childhood, as shown by some studies [3,17,19,20,21,22,23]. In most cases, arteriovenous malformations [AVMs] remain asymptomatic. However, AVMs can occur in the pulmonary, hepatic and cerebral circulations, and on certain occasions, they may lead to a wide range of life-threatening complications [7]. Interestingly, vasculature in HHT patients has been thought to undergo delayed vessel repair and prolonged inflammatory cell activation [24]. Even though, HHT is the second most common hereditary bleeding condition worldwide following von Willebrand disease in prevalence, there are no approved treatments for HHT-associated bleeding, and they are rarely described in the medical literature [25]. 

### 4.1. Epistaxis

Nosebleed is a very common manifestation of HHT but evaluated in very limited studies. According to the study by Ramakrishnan et al., 84% of patients with HHT, who presented with recurrent epistaxis, had a positive family history of HHT [6]. It is important for patients to treat nosebleeds in order to prevent their precarious consequences and improve their quality of life because it tends to be a huge social inconvenience and a source of embarrassment [26]. 

The first-line treatment for acute epistaxis is bidigital compression after nasal cleansing [27]. The expert panel also recommends the use of moisturizing topical therapies that humidify nasal mucosa (topical saline) and oral tranexamic acid. Clinicians may also consider ablative therapies such as laser treatment or radiofrequency ablation in case of failure of the previous methods [28]. Our patient was recommended to continue oral tranexamic acid supplementation after being discharged.

### 4.2. Visceral/Gastrointestinal Telangiectasias

Our patient was admitted to the hospital with acute upper gastrointestinal bleeding. The incidence of gastrointestinal teleangiectasias in patients with HHT is about 11 to 25% [29]. Gastrointestinal angiodysplasias are vascular malformations composed of dilated and tortuous arterial or venous capillaries, usually smaller than 5 mm in diameter, and located in the mucosal and submucosal layers of the gastrointestinal tract [30]. In the study by Sabbà et al., HHT2 patients seemed to show enhanced propensity to bleeding, although the underlying reason is still far from being elucidated [12]. Gastrointestinal telangiectasias is typically a late manifestation of HHT, usually occurring in the fifth decade or later, even though a few cases with a younger onset have been reported (including a few exceptional cases in pediatric age) [1,31,32,33,34,35]. 

The stomach and small intestine are more frequent sources of bleeding than the colon [29]. Extensive gastrointestinal bleeding or epistaxis may require blood transfusion, while some might even cause hemorrhagic shock [36]. Nevertheless, gastrointestinal hemorrhage, as well as nosebleeds, mainly contribute to iron deficiency anemia (IDA) as in our patient. According to the retrospective chart review of Pahl et al., 50% patients had medically proven anemia, of which it was mostly mild version [37]. The first-line approach is initiating iron supplementation, which has to be administered intravenously because, in this condition, the rate of blood loss exceeds the absorptive capacity [38]. According to worldwide recommendations [39,40,41], our patient received 200 mg iron sucrose injections once a day for 5 days. 

Nowadays, the Second International Guidelines for the Diagnosis and Treatment of HHT recommends using antiangiogenics and antifibrinolytics in bleeding [42]. The expert panel of the Second International Guidelines for the Diagnosis and Treatment of HHT recommends esophagogastroduodenoscopy as the first-line diagnostic test for suspected HHT-related bleeding [28]. While performing endoscopy of the stomach, telangiectatic structures are seen as similar in size and appearanc, to those in the mucous, often surrounded by an anemic halo. Less, commonly, AVMs and aneurysm may be visualized by gastrointestinal angiography. In order to cure anemia, the management of patients includes oral iron therapy and blood transfusion [19]. Furthermore, according to Chetcuti, pharmacotherapy such as somatostatin analogs can additionally help to improve transfusion requirements [43]. Our patient did not need to have a blood transfusion because her hemoglobin level was 9 mg/dl.

One of the most promising medications in HHT is bevacizumab, a recombinant, humanized VEGF-A (vascular endothelial growth factor-A) targeting monoclonal antibody, which was the first approved angiogenesis inhibitor. Initially approved for treatment of metastatic colorectal cancer with chemotherapy, its current indications are much broader, including metastatic breast cancer or ovarian cancer [44]. Bevacizumab is used off-label in the treatment of HHT-associated bleeding and anemia because its effectiveness has been only demonstrated in very few case reports and small cohort studies [45,46,47,48,49,50,51]. The international, multicenter study of intravenous bevacizumab for bleeding in HHT, the InHIBIT-Bleed study, has shown that a mentioned monoclonal antibody significantly reduced HHT-associated bleeding, causing the resolution of anemia in two-thirds of patients [25]. It seems to be very promising to use bevacizumab in HHT patients with severe gastrointestinal bleeding. It has effectively controlled severe GI bleeding in a patient with complicated HHT described by Ou et al. [52]. However, long-term side effects have not been studied yet among HHT patients [53]. 

Successive endoscopic treatments include monopolar and bipolar electrocoagulation, heater probe, injection sclerotherapy, Nd:YAG laser ablation and argon plasma coagulation. After that, many patients still require blood transfusions, although at a much reduced rate [54]. Our patient underwent argon plasma coagulation of the stomach lesions.

### 4.3. Hepatic AVMs

Telangiectasias can also occur in the liver where they occur as wide-spread vascular malformations (VMs). Given that the liver has a dual blood supply, three types of shunting are formed: arteriovenous (hepatic artery to hepatic vein), arterioportal (hepatic artery to portal vein) and portovenous (portal vein to hepatic vein). Despite all the types of shunting appearing concomitantly, one of them is in the ascendant and the predominant type of shunt may change over time [55,56]. Hepatic arteriovenous malformations (HAVMs) are rarely symptomatic [57,58,59]. 

Symptoms of hepatic involvement in HHT are abdominal pain, cholestasis, cholangitis, ascites, variceal hemorrhage and encephalopathy. Even though these symptoms occur only in 8% of patients, some of them may be life-threatening, and while diagnosing HHT, a hypothetical liver involvement should be taken into consideration [55,56]. HAVMs are significantly more frequent in patients who had the HHT2 genotype than in patients who had the HHT1 genotype [9,10,11,57,60,61,62]. The complications of liver involvement of HHT include high-output heart failure, portal hypertension and cirrhosis [63]. Abdominal Doppler ultrasound was used to exclude HAVMs in our patient. She did not present any HAVMs.

### 4.4. Pulmonary AVMs

Osler-Weber-Rendu syndrome also affects the respiratory system [64]. The most severe consequences of pulmonary arteriovenous malformations (PAVMs) are cerebral implications, with brain abscess being one of the most common and severe ones [65]. De Gussem et al. showed that the life expectancy of HHT patients systematically screened for HHT-related organ involvement and treated if needed was similar compared to their controls. However, untreated AVMs, especially in the lungs-pulmonary AVMs can result in morbidity with a decreased life expectancy [66]. PAVMs may be prevented by embolization therapy, and therefore it is crucial to diagnose them in advance [67]. Our patient underwent thoracoabdominal contrast computed tomography, which confirmed the absence of PAVM.

### 4.5. Cerebral AVMs and Spinal AVMs

Cerebral arteriovenous malformations [CAVMs] are more common in patients with HHT1 than HHT2 [9]. CAVMs affect approximately 10% of HHT patients. Due to the potentially devastating effects of cerebral hemorrhage, symptoms suggestive of CAVMs should be investigated with magnetic resonance imaging (MRI) [68]. Easey et al. showed that hemorrhages were six times more common in female HHT subjects [69]. Although most CAVMs are asymptomatic—they rarely bleed—the researchers also justify a more aggressive screening approach to identify small causative lesions amenable to treatment [68,69]. We performed MRI in our patient to exclude CAVMs. However, only leukoaraiosis was found. Spinal AVMs, which are substantially less frequent, usually present in early childhood with paralysis or complaint of back pain [68]. In the study by Brinjikji et al., paraplegia/paresis or quadriplegia/paresis was the presenting symptom in 57.7% of patients with spinal AVMs, respiratory distress occurred in 7.7% of patients and seizure occurred in 7.7% of patients [70]. 

## 5. Conclusions

It is worth noting that Osler-Weber-Rendu syndrome is an interdisciplinary condition. It is not only a gastrological issue but also a pulmonological, dermatological and neurological one [71]. Therefore, patients with upper gastrointestinal bleeding and other symptoms suggesting HHT should be referred to further genetic testing. A causal treatment is not available, and therefore, management of HHT patients is based on symptom relief [72]. The promising news comes from 20-year follow-up Danish study which suggests that among HHT patients, the risk of cancer is lower than in the healthy population [59]. Hence, since HHT is a multi-systemic disorder, we should be aware that a timely diagnosis of HHT in the presence of otherwise unexplained gastrointestinal bleeding is crucial in order to set up proper clinical-instrumental surveys of silent AVMs potentially affecting organs other than the GI tract and further prevent other complications. The diagnosis of HHT is a big challenge for doctors because of its rarity, and diagnosed HHT patients also suffer from improper medical management. The disease affects multiple organs, which requires multiple departments to be visited by HHT patients [73]. 

## Figures and Tables

**Figure 1 medicina-58-00333-f001:**
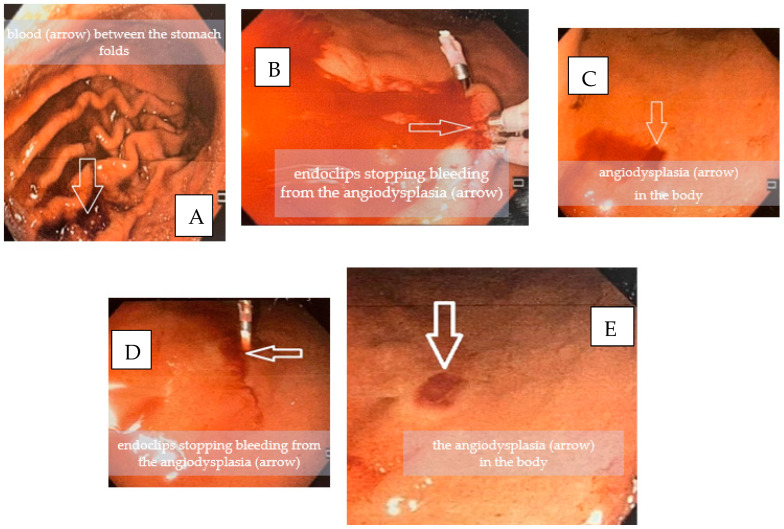
Gastroscopy images. (**A**)—blood between the folds of the stomach; (**B**,**D**)—endoclips; (**C**,**E**)—angiodysplasias of the body.

**Table 1 medicina-58-00333-t001:** Curaçao diagnostic criteria for HHT [5].

Symptoms	
1	Epistaxis—spontaneous and recurrent
2	Telangiectasias: multiple and characteristic sites (lips, mouth, fingers, nose)
3	Visceral/gastrointestinal telangiectasias, pulmonary AVMs ^1^, hepatic AVMs, cerebral AVMs and spinal AVMs
4	Family history: one first-degree relative

^1^ arteriovenous malformations.

**Table 2 medicina-58-00333-t002:** Laboratory test results of the patient with HHT.

Laboratory Test	Result	Normal Range
Hemoglobin	9.0 g/dL	11.5–16 g/dL
Hematocrit	29.4%	37–47%
Red blood count	3.91 T/L	3.5–5.2 T/L
Platelets	266/mm^3^	150–400/mm^3^
Leukocytes	3420/mm^3^	4000–10,000/mm^3^
Lymphocytes	11.8%	25–40%
Neutrophils	65%	50–62%
Alanine aminotransferase	14 U/L	<33 U/L
Asparate aminotransferase	16 U/L	<32 U/L
Total bilirubin	0.7 mg/dL	0.2–1.2 mg/dL
Creatinine	0.7 mg/dL	0.5–1.1 mg/dL
C-reactive protein	7.1 mg/L	0–5 mg/L
D-dimer	315 ng/mL	0–500 ng/mL
Sodium	138 mmol/L	136–145 mmol/L
Potassium	4.0 mmol/L	3.5–5.1 mmol/L
Iron	10 ug/dL	50–170 ug/dL
Ferritin	7 ng/mL	10–291 ng/mL
Vitamin B12	392 pg/mL	211–911 pg/mL

## Data Availability

All data are included in the main text.

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
