# Peer review of "A Rare Case of Upper Gastrointestinal Bleeding: Osler-Weber-Rendu Syndrome"

_medicina, 2022, doi:10.3390/medicina58030333_

Round 1
Reviewer 1 Report
review
A rare case of upper gastrointestinal bleeding: Osler-Weber-2 Rendu Syndrome
Abstract line 12
Furthermore, on certain occasions we may observe recurrent epi staxis, gastrointestinal bleeding and subsequent iron deficiency anemia.
This sentence is wrong and needs to be corrected, recurrent epistaxis is a very common sign of HHT present in around 90 % of patients, and not just present in certain occasions
You could write:
The most prominent symptom of HHT is epistaxis which together with GI-bleeding may cause iron deficiency anemia.
Line 20 abstract. What was the plan for the patient at discharge ?, it should include correction of anemia, screening for Pulmonary AVM and genetic testing including evaluation of 1.degree relatives
Introduction
Line 31
AVM of pancreas may occur, but the most important and severe complications should be mentioned first, Pulmonary AVM, Liver AVM and Brain AVM
Line 32-34
This sentence is not clear and needs rewriting
Line 34
This sentence is incorrect. Most HHT patients are not a-symptomatic, they suffer from epistaxis which may be social invalidating, they have cerebral abscess, and cerebral infarctions which can be prevented by screening and proper care. Due to lack of knowledge of the disease among health care professionals, many HHT patients are not diagnosed properly
Line 48
Needs re writing
None or one criterion makes the diagnosis unlikely change to: If None or only one criterion is met makes the diagnosis unlikely
Line 54
Less usually - This depend on your hospital and if asked for. Better write
The diagnosis may be confirmed through molecular testing, In this way HHT subtype can be identified, and 1.degree relatives can be tested for the disease.
Line 57
There is still discussion regarding the prevalence of GI-bleeding regarding HHT type 1 and type 2, and there is no conclusion, the citation used are wrong,
Figure 1
Picture 5 with two arrows, I believe they should remove the left arrow, as it does not point on a telangiectasia, more likely I points towards hematin.
Patient history line 90
Sentence Although, our patient 90 probably has a sporadic mutation leading to HHT due to a lack of history of hemoptysis, anemia or cutaneous disorders in her family, she was referred to genetic testing Is an incorrect conclusion and needs to be re-written, 1. Was there a family history of epistaxis, this is the most common symptom, 2. Hemoptysis is a very rare symptom of HHT 3 even if the patient has a sporadic mutation, knowning the HHT type is relevant for risk evaluation and for evaluation of 1.degree relatives especially her children
Discussion
Needs to be re-written
The authors should stick to the case and then discuss what to do with a case of newly diagnosed HHT with GI-bleeding, and why certain decisions are relevant, in terms of GI-bleeding and in terms of HHT.
Presently they try to make a review, but they do not focus
A case like this needs to be treated, as they describe, but if iron treatment and endoscopy is not enough the treatment include (steep wise) antifibrinolytics (tranexamic acid) and antiangiogenic treatment (Bevazicumab) So focus on the GI-bleeding treatment
Secondly and just as important a newly diagnosed HHT patient should:
Be offered mutation diagnostics so that family members can be evaluated and in order to calculate risk of different manifestations
Epistaxis is the epistaxis causing anemia, can she be treated (ref 26)
PAVM, she needs to be evaluated for presence of PAVM, as these untreated pose a considerable risk of cerebral insult or abscess ( ref 26)
Liver AVM This needs to be discussed with the patients and she needs screening if she has unexplained dyspnea, oedema especially if she carry an HHT type 2 mutation (ref 26)
Cerebral AVM, this needs to be discussed (as stated in reference 7)
Author Response
Dear Reviewer,
Thank you for your suggestions. We changed the abstract as you suggested. We improved patient history basing on your recommendations. We also re-wrote the discussion. We tried to focus on our patient and then showed how it is described by other researchers. We changed the structure of the discussion. We added more information about liver, cerebreal and spinal AVMs. We inserted information about recommendation to our patient to perform genetic testing which is impossible to perform in our Unit.
Thank you for your suggestions. Please see the attachment to see our improvements.
Best wishes

Reviewer 2 Report
Overall:
Jargielo et al report a case of an HHT patient presenting with acute upper gastrointestinal bleeding to a gastroenterology ward. Despite being quite an interesting case, Authors do not clearly evidence in what way this case report would add to the relevant literature, in terms of novelty, better knowledge of natural history of HHT-related GI involvement, complexity of healthcare, treatment options, etc.
At least some of these aspects should be underlined in the discussion section, or the case reported herewith would be of little value.
In addition, description of the HHT disease should be improved according to the relevant literature, and some references should be added to the bibliography section.
A possible suggestion is that this case might represent an HHT case displaying diagnostic delay, a phenomenon typically occurring in HHT (see Pierucci P, Lenato GM, Suppressa P, et al. A long diagnostic delay in patients with hereditary haemorrhagic telangiectasia: a questionnaire-based retrospective study, Orphanet J Rare Dis 2012; 7:33). Hence, since HHT is a multi-systemic disorder, Auhors may emphasize that clinicians should be aware that a timely diagnosis of HHT in presence of otherwise unexplained GI bleeding is crucial, in order to to set up proper clinical-instrumental survey of silent AVMs potentially affecting organs other than GI tract, and prevent other complications.
Specific remarks
- Abstract, Line 21
Although, Osler-Weber- Rendu syndrome is widely described in medical literature, there are only few articles presenting gastroenterological aspects or case reports
Actually, a non-negligible number of cases with GI bleeding have been reported in literature. Therefore, this statement should be modified, e.g. as follows:
Although, Osler-Weber- Rendu syndrome is widely described in medical literature, effective treatment of gastroenterological telangiectases are not always available and still lack standardization to date, which makes management of gastroenterological involvement still a challenging issue.
Introduction
- Page 1, Line 32
There is also an increased risk of pulmonary hypertension.
Although a few cases of pulmonary hypertension have been reported in HHT, such condition is recorded in only a small minority of patients, and it has no prominent relevance with GI involvement. Hence, I would suggest to delete it.
- Page 1, Line 32
In case of appearance of telangiectasias in gastrointestinal tract, bleeding and subsequent iron deficiency anemia are performed.
This is quite unclear. Maybe do Authors mean: “In case of onset of telangiectases in gastrointestinal tract, bleeding and subsequent iron deficiency anemia can arise, which require appropriate diagnostic work-up and clinical management” ?
- English language should be revised at many places.
e.g. the first sentence of the introduction reads now like this:
“Osler- Weber- Rendu disease, also known as hereditary hemorrhagic telangiectasia (HHT) is a rare, autosomal dominant disorder, with implication of the following gene mutations: ENG, ACVRL1/ALK1 and MADH4/SMAD4, characterized by multiple muco-cutaneous telangiectasias.”
Although the concept is basically right, and formal grammar is correct, wording is not completely understandable to the reader. I would suggest rephrase as follows: “Osler- Weber- Rendu disease, also known as hereditary hemorrhagic telangiectasia (HHT) is a rare, autosomal dominant disorder, characterized by multiple vascular lesions, ranging from muco-cutaneous telangiectases to larger arteriovenous malformations. Heterozygous pathogenic mutations in ENG, ACVRL1/ALK1 and MADH4/SMAD4 genes account for the disease”
Similarly, several other statements throughout the manuscript should reports concepts in a more precise and clear way.
- Page 1, Line 34
Even though there are many possible manifestations of this condition, the vast majority of patients with HHT is asymptomatic.
There is a fraction of patients with HHT who are asymptomatic, but it is not true that the vast majority of HHT patients is asymptomatic. Authors should report precise data reported in literature, in several genotype-phenotype correlation studies carried out to date, also according to age-dependant penetrance.
- Page 1, Line 35
Every case of gastrointestinal bleeding is a life-threatening emergency therefore it is crucial to establish the diagnosis to prevent possible complications, such as high-output heart failure, portal hypertension, liver failure, hemoptysis, polycythemia, cerebral abscess and stroke.
Several cases of gastrointestinal bleeding in HHT represent a life-threatening emergency indeed, but not all of them! Moreover, which is the link of GI bleeding with other HHT-related complications, such as liver failure, hemoptysis, polycythemia, cerebral abscess and stroke, which are secondary to liver AVMs or pulmonary AVMs, not to GI AVMs. Furthermore, high-output heart failure and portal hypertension are typical complication of HHT-related hepatic, rather than GI AVMS, although their presence can worsen risk of GI-bleeding-associated poor outcome. Maybe do Authors mean that assessing a proper diagnosis of HHT in presence of otherwise unexplained GI bleeding is crucial to set up proper clinical-instrumental survey of silent AVMs potentially affecting organs other than GI tract, leading to other complications?
- Page 1, Line 32
ACVRL1 mutations are more common in patients with gastrointestinal bleeding [9].
Most, but not all, studies carried out to date agree on this findings. However, they should be properly cited. The sentence should be better rephrased as follows:
HHT patients with gastrointestinal bleeding display more frequently a mutation in ACVRL1/ALK1 than in ENG gene, according to most genotype-phenotype correlation studies thus-far performed (also add the following references to the one reported one [9]: Sabba C, Pasculli G, Lenato GM, et al, Hereditary hemorrhagic telangiectasia: clinical features in ENG and ALK1 mutation carriers. Journal of Thrombosis & Haemostasis. 5(6):1149-57, 2007- Lesca G, Olivieri C, Burnichon N, et al, French-Italian-Rendu-Osler Network. Genotype-phenotype correlations in hereditary hemorrhagic telangiectasia: data from the French-Italian HHT network. Genetics in Medicine. 9(1):14-22, 2007 - van Tuyl SA, Letteboer TG, Rogge-Wolf C, et al, Assessment of intestinal vascular malformations in patients with hereditary hemorrhagic teleangiectasia and anemia. European Journal of Gastroenterology & Hepatology. 19(2):153-8, 2007 - Grève E, Moussata D, Gaudin JL, et al, High diagnostic and clinical impact of small-bowel capsule endoscopy in patients with hereditary hemorrhagic telangiectasia with overt digestive bleeding and/or severe anemia. Gastrointestinal Endoscopy. 71(4):760-7, 2010 - Canzonieri C, Centenara L, Ornati F, et al. Endoscopic evaluation of gastrointestinal tract in patients with hereditary hemorrhagic telangiectasia and correlation with their genotypes. Genetics in Medicine. 16(1):3-10, 2014 - Chetcuti Zammit S, Sanders DS, McAlindon ME, Sidhu R. The Impact of Small Bowel Endoscopy in Patients with Hereditary Hemorrhagic Telangiectasia. Turkish Journal of Haematology. 35(4):300-301, 2018).
Case report
- Page 2, Line 62
She had a history of recurrent epistaxis from the last many years. The patient had melena for 4 days which was confirmed in digital rectal examination. However, she also 64 received iron supplementation due to anemia.
The statement needs clarification. Please provide age of onset of epistaxis, albeit approximate (e.g. the from at least 10 years, or 20 yrs,…). The patient had been suffering from melena for 4 days (i.e. was she still having melena at admission?) or The patient had melena for 4 days (i.e. she had had melena for 4 days, but melena had stopped some time before admission?)? Moreover, how long had she been receiving melena? Did she ever have GI Bleeding episodes previously? History of transfusions?
Discussion
- Page 4, Line 105
This condition only becomes apparent in adulthood, with no age cut off and has numerous clinical representations [3,10,12].
This statement is not correct. Many of HHT-related manifestations do become apparent in adulthood, at a variable onset age, due to the age-dependent expressivity of the disease, but some manifestations can also occur in childhood, albeit at a lesser frequency, as shown by some studies (Gefen AM, et al, Asymptomatic pulmonary arteriovenous malformations in children with hereditary hemorrhagic telangiectasia. Pediatr Pulmonol. 2017, Giordano P, et al, Vasa. 2017 May;46(3):195-202 Hepatic angiodynamic profile in paediatric patients with hereditary haemorrhagic telangiectasia type 1 and type 2., Hosman AE, Screening children for pulmonary arteriovenous malformations: Evaluation of 18 years of experience. Pediatr Pulmonol. 2017 Sep;52(9):1206-1211, Gonzalez CD, et al, Epistaxis in children and adolescents with hereditary hemorrhagic telangiectasia. Laryngoscope. 2018 Jul;128(7):1714-1719).
Authors should also clarify that bleeding of GI telangiectases is typically a late manifestation of HHT, usually occurring in the 5th decade or later, even though a few cases with a younger onset have been reported (including a few exceptional cases in pediatric age), also adding all the relevant literature.
- The whole section could be shortened and should better focus on the comparison of the reported case with existing literature in HHT-related GI bleeding, emphasizing elements of novelty and/or complexity, as well as risk associated to delayed diagnosis in HHT-related GI and extra-GI manifestations. The references underlining the working group dedicated to HHT within the European Reference Network on Rare Multisystemic Vascular Diseases and the expert panel of the Second International Guidelines for the Diagnosis and Treatment of HHT are highly relevant and needs to be kept.
Author Response
Dear Reviewer,
Thank you very much for your suggestions. They were very helpful and interesting. We improved our manuscript basing on your recommendations. We attach the improved version. We added more references as you may see in the attachment.
Best wishes

Round 2
Reviewer 1 Report
Second review 29.1.2022
A rare case of upper gastrointestinal bleeding: Osler-Weber-2 Rendu Syndrome
The authors have improved the manuscript, there are however still some flaws that need to be corrected before publication can be considered.
Minor
The English needs to be reread some words a missing or wrong here and there line 46,50,140, 152,165
Line 83 did the patient have epistaxis since eleven years of age, or during the last eleven years ?
Rephrase line 214-218
The authors needs to go through the reference list thoroughly
Some of the references are listed twice
Major
The case story has been improved, but the authors does not mention if the patient has telangiectatic lesions at characteristic sites (skin of the face, oral mucosa, nasal mucosa, lips) So I cant tell if she full fil the two (GI-bleeding + epistaxis) or three Curacao criteria (Telangiectatic lesions +GI-bleeding + epistaxis) be aware that telangiectatic lesions in GI tract does not count as characteristic site.
The authors still state that GI bleeding is more common in HHT2 Than HHT 1, they have put more references on the statement (9-15) which they have not read, since some of them do not mention genotype relation and other state opposite correlation,
9 have not made investigation them self only refers to others
10 state No significant difference in GI bleeding between HHT1 and HHT2
11 state that HHT 2 patients have more common bleeding than HHT 1 So this is an OK reference (but the only one supporting the authors statement)
12 state the opposite HHT 1 patients are more severely affected with GI bleeding than HHT2
13 state they find no Correlation with GI bleeding and Genotype
14 state HHT 1 patients are more severely affected with GI bleeding than HHT2
15 does not mention Genotype
Further the authors state that spinal AVM are seen only in HHT 2 This is a wrong statement and a wrong citation
The rather old reference identifying 2 spinal AVM in a small population of HHT type 2 patients
A quick pub med search identify a paper from 2016 (Epub 2016 Jan 27.
Spinal arteriovenous fistulae in patients with hereditary hemorrhagic telangiectasia: A case report and systematic review of the literature
Waleed Brinjikji 1, Deena M Nasr 2, Harry J Cloft 3, Vivek N Iyer 4, Giuseppe Lanzino 5)
with 25 HHT case reports with spinal AVM, only in 6 patients mutation diagnostics had been performed however 2 case had HHT 1 and 4 HHT2
Conclusion line 292-297 is not relevant to put in the conclusion but belong in the case description
Reviewer 2 Report
The manuscript has been improved. However, many sections are still too long and needs shortening. Furthermore, some concepts needs to be refined. Bibliography is now appropriate.
Specific remarks:
Abstract
Line 21
She was referred to further genetic testing including evaluation of first degree relatives.
Line 24
Due to the fact that patient did not manifest any other symptoms of HHT, the “watch- and-wait strategy” was applied
When no symptoms are evident, but occult AVMs are present, watch- and-wait strategy is not the best management option. Authors state that no AVMs were found at instrumental screening, which is the main reason why watch- and-wait strategy is justified. Therefore, the sentence should be changed to:
Due to the fact that the patient did not manifest any other HHT-related symptoms of HHT, and that the instrumental screening disclose no silent AVMs in other organs, the “watch- and-wait strategy” was applied
Introduction
Line 41
Interestingly, epistaxis is more common to be secondary to telangiectasia of the nasal mucosa among patients with HHT.
This is an obvious statement. It is of little value. Authors should change as follows:
However, the most frequent HHT-related manifestation is represented by repeated epistaxis episodes, secondary to telangiectasia of the nasal mucosa.
Line 43
Furthermore, patients with HHT can have cerebral abscess, cerebral infarctions which could be prevented by screening and proper care
English language is not correct. Should read: “Furthermore, patients with HHT can suffer from cerebral abscess and/or cerebral infarctions, which could be prevented by screening and proper care”
HHT diagnosis
Line 66
In this way HHT subtype can be identified and 1. degree relatives can be tested for the disease.
1st degree, not 1. degree relatives
Line 71
HHT patients with gastrointestinal bleeding display more frequently a mutation in ACVRL1/ALK1 than in ENG gene
ACVRL1 and ALK1/ACVRL1 are two equivalent wording to define the same gene, in current human genetics nomenclature. However, if you chose to employ “ACVRL1” in the previous sentence, then you should go on with ACVRL1, and not with the ACVRL1/ALK1 wording.
Line 108
family history of epistaxis or any other potential HHT syndromes is unknown so was referred to further genetic testing including evaluation of 1st degree relatives.
Genetic evaluation of 1st degree relatives can only be performed after the mutation has been identified in the proband, which is not the case. The sentence should be change as follows: Family history of epistaxis or any other potential HHT manifestation is unknown. In order to identify the underlying genetic mutation and enroll 1st degree relatives to further genetic testing, the patient was referred to mutational analysis of the HHT-causing genes.
Line 110
Due to the fact that patient did not manifest any other symptoms of HHT, the 110 “watch-and-wait strategy” was applied.
Same criticism as Abstract Line 24
Line 111
Our patient may represent an HHT case displaying diagnostic delay, a phenomenon typically occurring in Osler-Weber-Rendu syndrome [16].
The statement is correct, nut needs an explanation: should be changed as follows: Our patient seems to represent an HHT case displaying diagnostic delay, a phenomenon typically occurring in Osler-Weber-Rendu syndrome [16], as the diagnosis was established after 11 years of unexplained nosebleeds.
Discussion
Discussion has been improved, if compared to the initial version of the manuscript. However, it is still too long and needs shortening at many places.
Line 126
Our patient probably represents a sporadic mutation leading to HHT due to a lack of history of hemoptysis, anemia or cutaneous disorders in her family.
This statement cannot be considered valid unless mutational analysis has not been concluded. Delete.
Line 136
Interestingly, HHT patients experience persistent inflammation [24].
That statement is not fully correct. Interestingly, vasculature in HHT patients has been thought to undergo delayed vessel repair and prolonged inflammatory cell activation [24].
Line 162
In the study by Sabbà et al., there was no significant difference in GI bleeding between HHT1 and HHT2 group (60% vs. 51%) [12].
This study actually reported no significant difference in GI telangiectases between HHT1 and HHT2 group, but HHT2 patients seemed to show enhanced propensity to bleeding, although the underlying reason is still far from being elucidated.
Line 202
Furthermore, in the Australian study there was no 202 significant effect of intranasal bevacizumab used in the treatment of HHT-related epi-203 staxis [52] while Norwegian study showed the improvement of quality of life in HHT 204 patients when using intranasal bevacizumab [53].
Why adding a sentence regarding intranasal bevacizumab in a section devoted to GI bleeding? It is completely redundant and should be eliminated.
Liver
In the study by Karlsson et al., one of the HHT2 patients developed HAVM-associated terminal 223 liver failure and was subjected to a liver transplantation at the age of 46 years. He was 224 alive 14 months after the transplantation [60].
The immediate goals of treating HAVMs are to alleviate symptoms and to stabilize hemodynamic condition [9].
Since the reported case has no HAVMs, these sentences are redundant, in my view.
Lines 257-262
As patients with HHT are at increased risk for both bleeding and clotting events, …
Did the patient suffer from anticoagulant intolerance? It is not reported. Hence, it seems completely redundant and should be removed.
Line 262
After a six-month control of the patient's situation, it has been found that she has experienced 263 two episodes of epistaxis treated symptomatically (tranexamic acid 500 mg 3x/day).
This sentence should be better placed in the Epistaxis section.
Author Response
Dear Reviewer, Thank you for your amendments. We applied for all of the corrections and I'm sending a new version of our manuscript. However, I have a few questions regarding some of your notes. The sentence "After a six-month control of the patient's situation, it has been found that she has experienced two episodes of epistaxis treated symptomatically (tranexamic acid 500 mg 3x/day)." was already transferred from section epistaxis (discussion) to case report, as this was a request from the other reviewer. We also think that this sentence is an important part of the description of the case report. Therefore, may we kindly ask you to leave this sentence in the case report section or do you still prefer to move it? We shortened discussion according to your notes and we would like to ask if this section still needs to be shortened? If yes, we thought about deleting the whole part about bevacizumab as our patient did not receive this kind of treatment. Or maybe because we have already deleted the last sentences in this bavacizumab chapter, we can leave this part as it is a promising possibility of treatment? Thank you for your helpful tips. Yours faithfully, authorsRound 3
Reviewer 1 Report
Most improvements have been made
There are still some wording and spelling mistakes
I am sorry you need some more corrections before you are finish
- The last sentence in the conclusion (that the patient received TXA) was moved, but not to the case story instead to the discussion line 262-264 needs to be moved to line 112
- You cited the inHIBIT-bleed manuscript, line 199, but reference 44 as cited is not the inHIBIT-bleed , the inHIBIT-bleed reference is not in your list
- lin 248 your write (6.18;2.227 to13.45) this does not make any sense
- line 249 you write CAVM never bleed, rarely bleed would be more correct
Author Response
Dear Reviewer,
We attach the file. Thank you for your suggestions. We also improved our English.
Best wishes,
authors
